



# Demonstration of Offshore Wind Integration with an MMC Test Bench featuring Power-Hardware-in-the-Loop Simulation

Fisnik Loku[1], Philipp Ruffing[1], Christina Brantl[1], Ralf Puffer[1]

[1]RWTH Aachen University, Aachen, 52056, Germany

*Correspondence to*: Fisnik Loku (f.loku@iaew.rwth-aachen.de)

**Abstract.** The integration of offshore wind energy into the existing power system is continuously growing. With the increasing distance of the offshore wind farms (OWF) to the onshore AC transmission systems, HVDC systems are emerging as a preferable solution for the connection of OWF due to their techno-economic advantages in comparison to AC subsea connections. Integrating HVDC systems into the existing AC systems poses various planning and technological challenges.

To be able to overcome these challenges a variety of studies has to be conducted, e.g. the HVDC system behaviour under faults. Simulations using electromagnetic transient (EMT) tools represent a generally accepted method to conduct the relevant studies. To increase the trust in the developed concepts subsequent hardware demonstrations would be preferable. However, performing these investigations with full-scale components is often not an option due to unavailability and high costs. As an alternative way, Power-Hardware-in-the-Loop (PHiL) approaches are considered. In this context, a new and worldwide unique

laboratory demonstrator - the MMC Test Bench - is set up at RWTH Aachen University as part of the Horizon2020 project PROMOTioN. Here, laboratory-scaled Modular Multilevel Converters (MMCs) are used, which are connected on the DC side by cascaded Pi-line segments. The adjacent AC grids (i.e. offshore wind farms, AC transmission networks) are represented by real-time simulators (RTS) and connected to the MMCs via high bandwidth linear power amplifiers (PA).

In this work, the MMC Test Bench is initially described. Afterwards, the PHiL set-up to demonstrate the implemented controls

for an OWF connected to shore via an HVDC link is explained. To allow the joint operation of the hardware set-up and the RTS in a stable manner, adequate PHIL interfaces algorithms have to be designed and the scaling between the RTS, the power amplifiers and the hardware is explained. The connection of the OWF represents a special challenge for PHiL demonstrations as the OWF represents a weak AC system with the MMC in grid forming mode. In a next step, the results of the successful demonstration of the interconnection of the OWF via an HVDC link with the MMC Test Bench are presented. The system

behaviour in stationary and transient operation is analysed based on the wind farm start-up sequence as well as different cases of wind infeed fluctuations. The results are compared to a simulated full-scale model and deviations are discussed.

## 1 Introduction

Offshore wind capacity is expected to amount to 70-100 GW in Europe by 2030 (WindEurope, 2017). As the distance between the OWF to the onshore transmission system is increasing and AC cable connections are limited in length due to the need for





reactive power compensation HVDC systems are increasingly used as a connection of the OWF to shore. To increase the availability of the connection of OWFs to shore, it is envisaged to use multi-terminal DC (MTDC) networks instead of single point-to-point connections for future OWF connections (PROMOTioN WP12, 2017). However, a number of technical challenges needs to be overcome, before MTDC networks can be implemented. EMT simulation software is a generally accepted tool to conduct investigations regarding the system behaviour in stationary and transient operation and they are

essential in the development of various network topologies and control algorithms. However, in order to increase the technology readiness level (TRL), the developed control algorithms and the various network topologies need to be demonstrated using hardware components (P. Rault and O. Despouys, 2018). Demonstration at full-scale is limited to very specific cases due to several reasons such as high costs and limited access to infrastructure. Power-Hardware-in-the-Loop (PHiL) systems on the other hard are considered as a practical and efficient alternative for the demonstration purposes (Brandl,

40    2018).

This paper presents the implementation of an HVDC-connected OWF connected to an AC grid, using real hardware MMC components of the MMC Test Bench, a PHiL system set up at the RWTH Aachen University. For this purpose, a point-to-point connection of an OWF in a symmetrical monopole network configuration is considered. Initially, the investigated network configuration as well as the MMCs and the OWFs along with their respective control algorithms are introduced. Next,

the PHiL implementation of the OWF is described in detail. Test cases including various operating points of the MTDC system are conducted for the full-scale simulation model and the MMC Test Bench. Finally, the responses of the simulated full-scale system and the MMC Test Bench are evaluated and the obtained results are compared.

**Modular Multilevel Converters (MMCs)**

In HVDC transmission systems, MMCs are the state-of-the-art voltage source converters (VSCs). The MMCs consists of identical submodules (SM) that are connected in series in a phase-arm. For a three phase-system, an MMC consists of six phase-arms, with two phase-arms forming a phase-leg. Furthermore, each phase-arm has an inductor in serial connection to the SM. The inductor prevents potentially high transient currents in the phase-arms and is needed for the control development of the MMC control. Each SM comprises a capacitor and by using bipolar switching devices, each SM can be controlled to

insert or bypass the capacitor. This way, each phase-arm can be controlled to operate as a voltage source and therefore provide the desired AC and DC voltages of the MMC. The SM can be configured as either half- or full-bridge submodules (Sharifabadi et al., 2016). Figure 1 shows the schematic design of an MMC station.



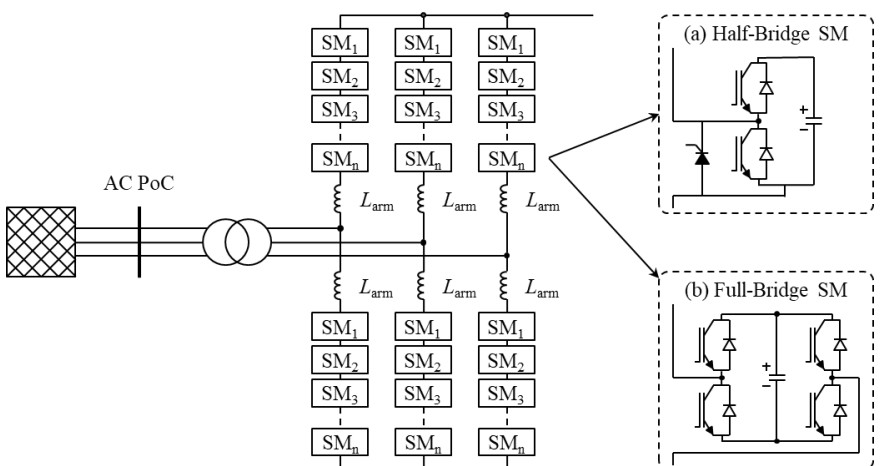

**Figure 1. Schematic design of an MMC station with (a) half-bridge and (b) full-bridge submodules (Ruffing et al., 2018)**

### Control of the MMCs

The implemented MMC control is based on Cigre brochure 604 "Guide for the Development of Models for HVDC Converters in a HVDC Grid" (CIGRÉ and Comité d'études B4, 2014, p.4). A cascaded vector control is used, as shown in Fig. 2. The dispatch controller provides the reference control values such as the reference active and reactive power, the AC and the DC voltage to the upper level control. For a non-islanded control, at the upper level control, the active power or the DC voltage and the reactive power or the AC voltage are controlled to their reference values that are provided by the dispatch controller and consequently provide the reference current values in the direct (d) or quadrature (q) components, respectively. The measured d and q components of the AC current are then controlled to their reference values. For the AC current control, the decoupled double synchronous reference frame is used (Teodorescu et al., 2011). Furthermore, the MMC can act as a grid reference setting the AC voltage and frequency at its terminals (Jing et al., 2018). This mode is called islanded mode, as it serves to connect weak or islanded AC grids such as OWF. At the lower level-control, the converter controls, such as the circulating current control and the energy balancing control as well as the valve control, such as the capacitor balancing and the modulation take place (Hahn et al., 2016).

### Offshore Wind Farms

For the OWF, a full-scale converter average model and a synchronous generator model are used. Equivalent voltage sources are used to represent the voltage source converters of the OWF. This is assumed as sufficient to investigate the control system dynamics and the system behaviour. The OWF is modelled as an aggregated model and is rated at $P_{\text{OWF,nom}} = 400$ MW. The OWF is connected to a 25-kV collector grid that exports power to a 155-kV grid. The generator-side VSC controls the turbine speed and the grid-side converter controls the DC voltage and the reactive power that are provided to the collector grid (Hansen and Margaris, 2014).



**Control of the OWFs**

Figure 2 shows the OWF control that is used for the investigations in this paper. The OWF control contains of two parts, the

inverter control and the synchronous machine control. Similar to the MMC control, the inverter control is a cascaded vector control (Mathworks, 2019), (Miller et al., 2003). Here, the dispatch controller provides the reference values of the DC voltage and reactive power to the OWF control. In addition, a wind speed reference is set in the model to emulate the existing wind conditions. The upper-level controls control the DC voltage to its reference value to provide the reference d component of the AC current. The reactive power is controlled to provide the reference AC voltage value, to which the AC voltage is controlled

and subsequently the reference q component of the AC current is obtained (Hansen and Margaris, 2014). A current controller controls the AC current to its previously determined reference values and converts them into reference arm voltage signals. The modulation algorithm in the lower-level controls converts the provided reference arm voltages to switching pulses which are used to control the inverters. The synchronous machine controls consist of the pitch controller and the rotor speed regulator. The pitch angle controlled by the pitch controller in combination with the reference wind provides the torque of the wind

turbine. The mechanical power for the synchronous machine shaft results from the torque of the wind turbine and the generator speed.

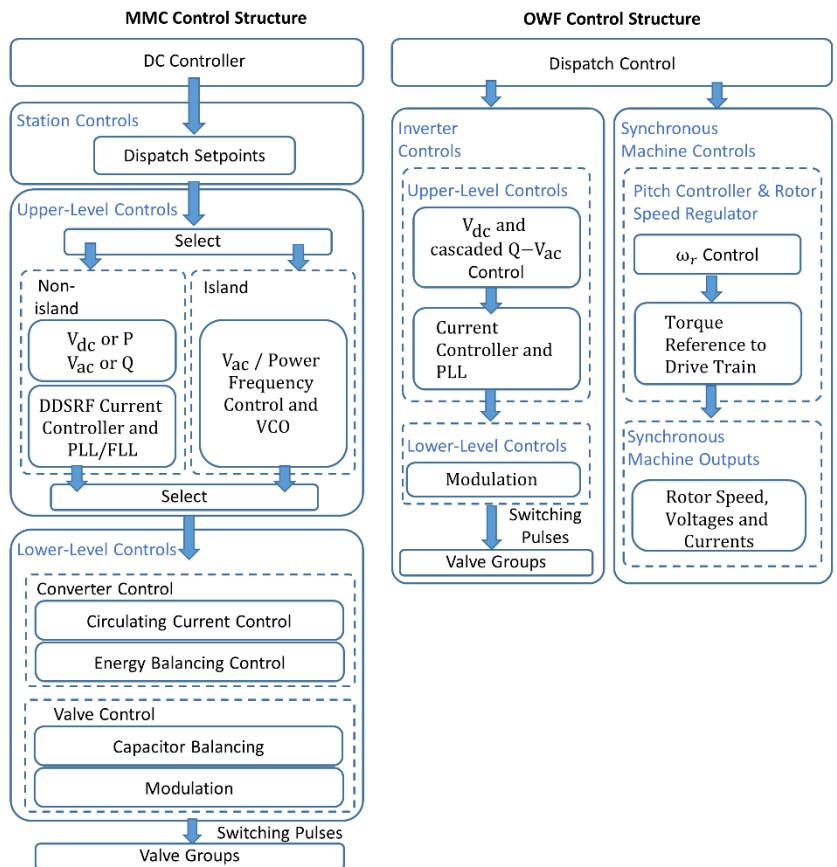

**Figure 2. Control hierarchy of the MMCs (left) (CIGRÉ and Comité d'études B4, 2014) and the OWFs (right)**



## 2 MMC Test Bench

The MMC Test Bench is a laboratory-scaled HVDC demonstrator consisting of laboratory-scaled components and real-time simulators (RTS) (OPAL-RT, 2019). The MMC Test Bench can be mainly separated into the AC systems (e.g. OWFs and AC transmission networks) and a DC grid. As shown in Fig. 3, the AC systems are simulated in Matlab/Simulink and RT-Lab and implemented in an RTS, i.e. an OPAL-RT OP5707, which in combination with the four linear PAs can emulate the respective simulated AC system. The PAs can either be configured in voltage source mode or current source mode. The choice depends

on the control of the connected systems. The DC grid of the MMC Test Bench consists of eight laboratory-scaled MMC stations. Furthermore, each MMC station includes an RTS, i.e. OPAL-RT OP4510, where the MMC control algorithm is implemented. Here, the upper level control of the MMC is implemented on the Central Processing Unit (CPU) of the RTS and the lower level controls are implemented on the Field Programmable Gate Array (FPGA) of the RTS. The dispatch controller of the MMCs and the supplementary control levels are implemented on a separate OP4510 RTS.

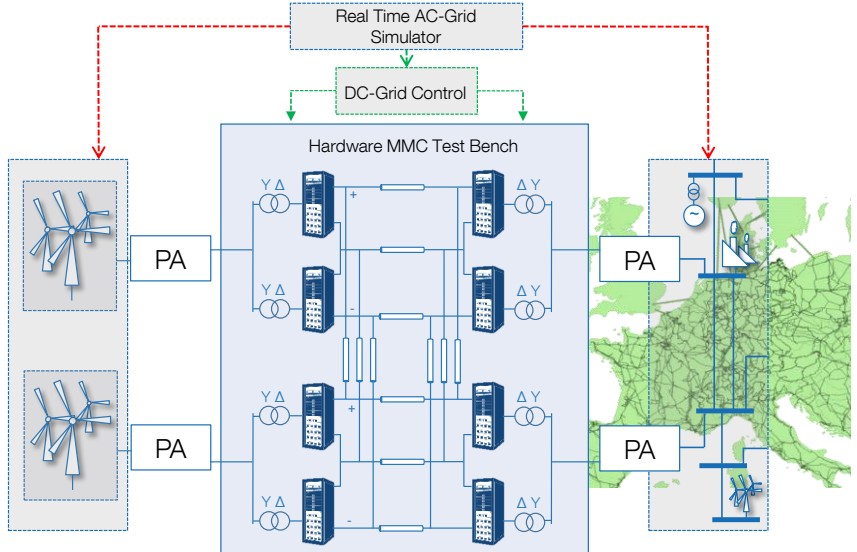


**Figure 3. Schematic design of the MMC Test Bench (PROMOTioN, 2019)**

The MMC ratings are presented in Table 1.

**Table 1 – MMC Ratings**

| Nominal DC Voltage | $V_{DC,n}$ | 400 V |
|---|---|---|
| Nominal DC Current | $I_{DC,n}$ | 15 A |
| Nominal Output Power | $P_{DC,n}$ | 6 kW |
| Nominal AC Voltage per phase | $V_{AC,1}/V_{AC,2}$ | 400/208 V (3ph-LL) |
| Nominal AC RMS current | $I_{AC}$ | 16.7 A |
| MOSFET Switching Frequency | $f_{sw}$ | 0-10 kHz |



Cascaded Pi-sections are used to emulate the DC line connection between the MMCs. Depending on the network configuration, a single Pi-section can emulate 50 km of a DC cable length for a symmetric monopole configuration and 25 km for a bipolar configuration. The Pi section ratings are shown in Table 2.

**Table 2 – Pi-section Ratings**

| Pi section resistance | $R_{PI}$ | 0.037 Ω |
|---|---|---|
| Pi section inductance | $L_{PI}$ | 10 mH |
| Pi section capacitance | $C_{PI}$ | 100 μF |

## 3 Investigated Network

The investigated network is a symmetric monopole point-to-point configuration and it consists of two MMC stations, as shown in Fig. 4. The network is initially implemented as an offline simulated model with the MMCs rated at full-scale at $P_{MMC,nom}$ = 1200 MW active power and $V_{DC,nom}$ = ± 320 kV DC voltage. The network configuration is then implemented using the MMC Test Bench with the lab-scale rated MMCs. For comparison reasons between the simulated full-scale model and the MMC Test Bench, the MMC reference control values are going to be referred to in p.u. system.

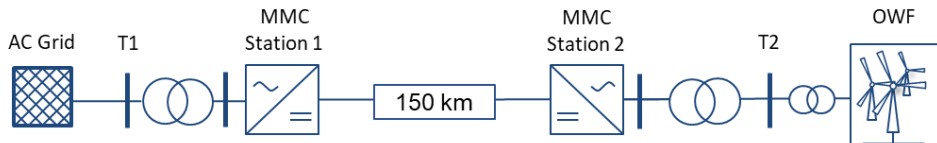


**Figure 4. Schematic view of the investigated network**

The AC grid model is rated at $S_{nom}$ = 30 GVA apparent power and at a rated phase-to-phase AC voltage of $V_{AC}$ = 400 kV at $f$ = 50 Hz frequency. MMC station 1 is connected to the AC grid and it controls the DC voltage to $V_{DC,n}$ = 1 p.u.

The OWFs are simulated as a synchronous generator and full-scale converter (Type 4) average model, where the voltage-130    sourced converters are represented by equivalent voltage sources, which is sufficient for investigations regarding control system dynamics and system interaction (Mathworks, 2019), (Miller et al., 2003). MMC station 2 is connected to the OWF and it is in grid forming control mode. It controls the AC voltage to $V_{AC,LL,RMS}$ = 1 p.u.

## 4 Power Hardware in the Loop Implementation

In this section, a general introduction to PHiL is initially given and then the PHiL implementation of the investigated system 135    is described.

To couple a simulated network with hardware components, several PHiL interfaces, such as the Ideal Transformer Method (ITM) or the Damping Impedance method (DIM) can be applied (Ren et al., 2007). For the investigations in this paper, ITM





is applied as it is an interface algorithm with a high accuracy (Ren et al., 2007). ITM can be implemented as a voltage or a current type and this is shown in Fig. 5. The voltage type ITM is typically used for a network consisting a simulated voltage

source and a hardware component. Here, the simulated voltage is measured and it is provided as a reference voltage to a controlled voltage source (CVS) on the hardware side of the network. As a result, a current flow on the hardware side of the network and this current is measured and provided to a simulated controlled current source (CCS). For the current type ITM, the current is measured at the simulated side of the network and provided to a CCS on the hardware side of the network. Here, the voltage is measured and is provided as a reference signal to the CVS on the simulated side of the network.

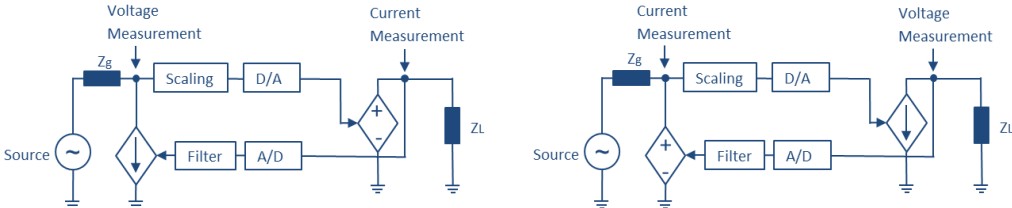


**Figure 5. ITM interface algorithm – voltage type (left), current type (right) (Ren et al., 2007)**

A challenge for PHiL investigations is the design of the interface algorithm between the hardware components and the RTS due to the instabilities that can occur when the whole system is considered. These instabilities can be for instance the measurement noises that can influence the measured voltage or current on the simulation side of the circuit, which then can be

transferred and amplified to the hardware side of the network. The overall stability also depends on the strength of the simulated grid, as a strong grid is normally less susceptible to the effects of the measurement noise. The transfer functions of the above-described ITM algorithms are presented in Eq. (1) and Eq. (2), respectively (Ren et al., 2007):

$$G_{ITM_V} = -\frac{Z_g}{Z_L} \cdot e^{-s\Delta t} ,\qquad(1)$$

$$G_{ITM_I} = -\frac{Z_L}{Z_g} \cdot e^{-s\Delta t} ,\qquad(2)$$

where, $\Delta t$ is the time delay of the interface. $Z_L$ and $Z_g$ are the equivalent impedances in the hardware circuit and the simulated network, respectively. The indexes $V$ and $I$ represent the transfer function of the voltage and current type, respectively. Depending on the relation of the equivalent impedance of the simulated network $Z_g$ and the equivalent impedance of the hardware network $Z_L$, the interface algorithms can be stable or not. Using the Nyquist Criterion, an impedance ratio greater than one would mean that the interface algorithm is not stable and a value of smaller than one would mean a stable interface

algorithm. For the MMC Test Bench, the AC grid is the same grid that is used for the full-scale model and is simulated as a strong grid. Therefore, no PHiL interface is used for the connection of the AC grid to the MMC Test Bench in this paper. Here, the AC voltage values of the simulated AC grid are measured and are scaled down, such that the line-to-line voltage of a single phase $V_{LL} = 400$ kV is equal to an analogues reference value of 10 V given to the PA. The PA is calibrated such that for a 10 V analogue signal, it provides a $V_{LL} = 400$ V to MMC station 1. MMC station 2 is defined to be in islanded control mode and

provides AC voltage to the OWF. As the voltage source is on the hardware side of the circuit, the current type ITM is used in



order to couple the OWF with MMC station 2. Furthermore, the OWF is rated at $P_{OWF,nom} = 400$ MW and the MMC station 2 at $P_{MMC,nom} = 1200$ MW. This means that the Thevenin equivalent impedance of the OWF will be higher than the equivalent impedance of the MMC station, and therefore the ITM interface algorithm is stable. Here, as shown in Fig. 6, the voltage provided by MMC station 2 is measured, filtered in order to remove any unwanted measurement noise, scaled up and provided

to a simulated CVS as a reference voltage. The reference voltage is scaled up according to Eq. (3):

$$V_{ref} = \frac{155\ kV}{400\ V} \cdot V_{meas} , \qquad (3)$$

with:

$V_{meas}$ – being the measured voltage at the hardware side of the network.

$V_{ref}$ – being the reference voltage provided to the simulated CVS.


To enable a stable start-up of the system, the control of the OWF is initially disabled until the reference voltage is provided to the CVS. As soon as the CVS receives the reference AC voltage from the MMC, the control of the OWF is enabled and the OWF starts to inject current into the grid. This current is measured, scaled down and provided to the PA as a reference current, which is in current mode and acts as a CCS. The reference current is scaled up according to Eq. (4):

$$I_{ref} = \frac{6\ kW}{1200\ MW} \cdot \frac{155\ kV}{400\ V} \cdot I_{meas} , \qquad (4)$$

with:

$I_{ref}$ – being the reference current provided to the PA.

$I_{meas}$ – being the measured current at the simulated side of the network.

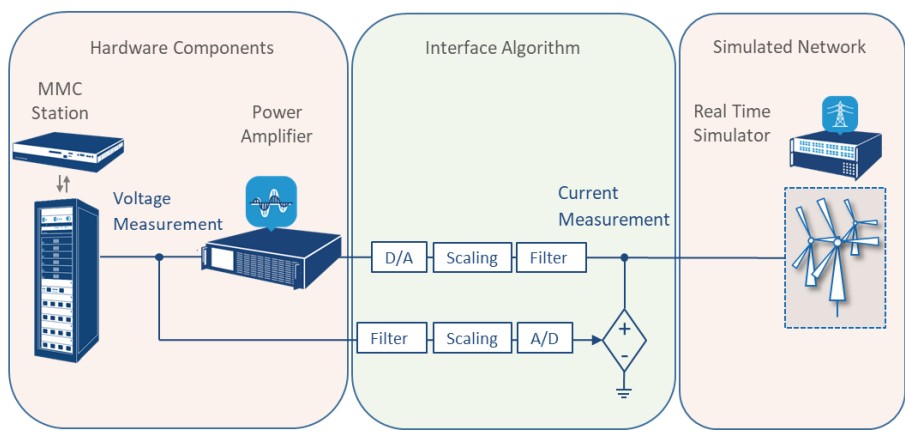

**Figure 6. PHiL implementation of the investigated MTDC network**



## 5 Investigated Test Case

To allow a stable start-up of the system, the system is energised from the onshore side. The simulated AC grid provides the nominal AC voltage as a reference value to its connected power amplifier, such that the power amplifier provides an AC voltage of 1 p.u to the hardware MMC. Consequently, MMC station 1 charges its submodules and controls the DC voltage to $V_{DC} = 1$ p.u. While MMC station 1 provides the DC voltage, MMC station 2 charges from the DC side and afterwards the grid forming control is activated. Subsequently, the OWF control is enabled. The active power transfer from the OWF to the DC grid is determined by the wind speed. To show the stable operation of the system for wind power fluctuations, the wind speed changes according to Table 3 are considered.

**Table 3 - Wind speed set point changes for OWF**

| TEST CASE | OWF | | |
|---|---|---|---|
| Set-point change time | $t_1 = 10$ s | $t_2 = 70$ s | $t_3 = 130$ s |
| From [m/s] | 5 | 15 | 12 |
| To [m/s] | 15 | 12 | 0 |

This test case is applied to both the full-scale simulated model and the MMC Test Bench. The results for both models are compared and presented in the following and the differences are investigated.

## 6 Results

To investigate the behaviour of the system for the simulated models and the MMC Test bench, the AC voltage, the DC voltage and current as well as the active power is compared.

Figure 7 shows the RMS value of the AC voltage measured at the terminals of the MMCs. At terminal 1, the RMS value of the voltage measured for the full-scale model and the MMC Test Bench shows very close values. The difference between the RMS values of the AC voltage for the full-scale model and the MMC Test Bench is ~ 0.005 p.u. This difference is due to the fact, that for the MMC Test Bench, the simulated AC grid values are provided to the hardware system via the PA. Even though the PA has been specially calibrated, it has an error margin and the ~ 0.005 p.u. result from this error margin and has to be accounted for in the analysis of the results. For the purpose of this paper this error is not deemed critical.

At terminal 2, a similar behaviour as for terminal 1 can be observed. Here, the main difference to terminal 1 is the fact that the AC voltage is not simulated as a strong grid, but it is set by MMC station 2. For the simulated full-scale model, the RMS value of the AC voltage is $V_{RMS,LS} = 1.002$ p.u. and thus higher than at terminal 1 due to the islanded control mode. Further investigations regarding this difference are not presented in this paper, as this difference is considered acceptable. Similar to





terminal 1, the AC voltage for the MMC Test Bench shows a difference of ~ 0.004 p.u. compared to the simulated full-scale model due to the accuracy of the PA and is considered as acceptable for the investigations as well.

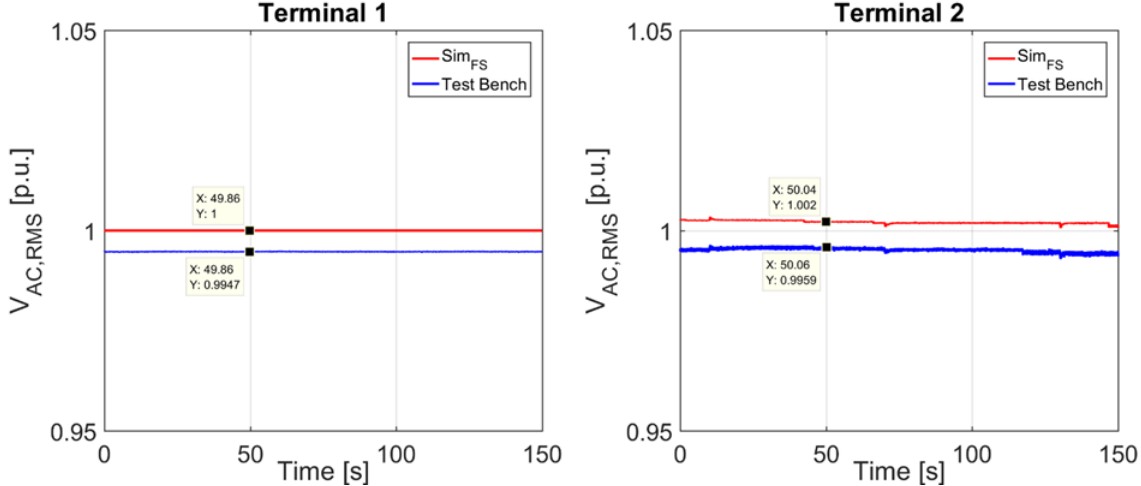

**Figure 7: AC Voltage RMS value measured at terminal 1 (left) and at terminal 2 (right)**

Furthermore, for terminal 2, the MMC Test Bench shows higher ripples than the full-scale model. This is due to the fact that compared to the full-scale MMC model with 350 SM per phase-arm, the hardware MMC components of the MMC Test Bench consist of only 10 SM per phase-arm, which limits the quality of the provided AC voltage.

Figure 8 shows the DC voltage measured at the MMC stations. MMC station 1 controls the DC voltage to $V_{DC,MMC1} = 1$ p.u.
Here, it can be seen that there is no considerable difference between the models. The MMC Test Bench shows higher ripples than the simulated full-scale model, due to the same reasons as for the AC voltage comparison. However, the simulated full-scale model and the MMC Test Bench have the same DC voltage level. Additionally, to the DC voltage level of 1 p.u., the simulated full-scale model and the MMC Test Bench show the same DC voltage behaviour when the power infeed set-points at MMC station 2 change. At MMC station 2, the DC voltage is initially $V_{DC,MMC2} = 1.001$ p.u. At 10 s, the wind speed changes
from $v_{wind} = 10$ m/s to $v_{wind} = 15$ m/s. The DC voltage of the simulated model and the MMC Test Bench increases at terminal 2, as the power infeed to the DC grid from the OWF increases. However, the MMC Test Bench shows higher DC voltage level and the same is valid for if the wind speed is reduced from $v_{wind} = 15$ m/s to $v_{wind} = 12$ m/s. For lower wind speeds however, the MMC Test Bench shows lower DC voltage levels. Overall, this difference is less than ~0.01 p.u. and is considered as acceptable for the investigations in this paper, as the behaviour of the DC voltage at MMC station 2 is almost identical for the
full-scale model and the MMC Test Bench.

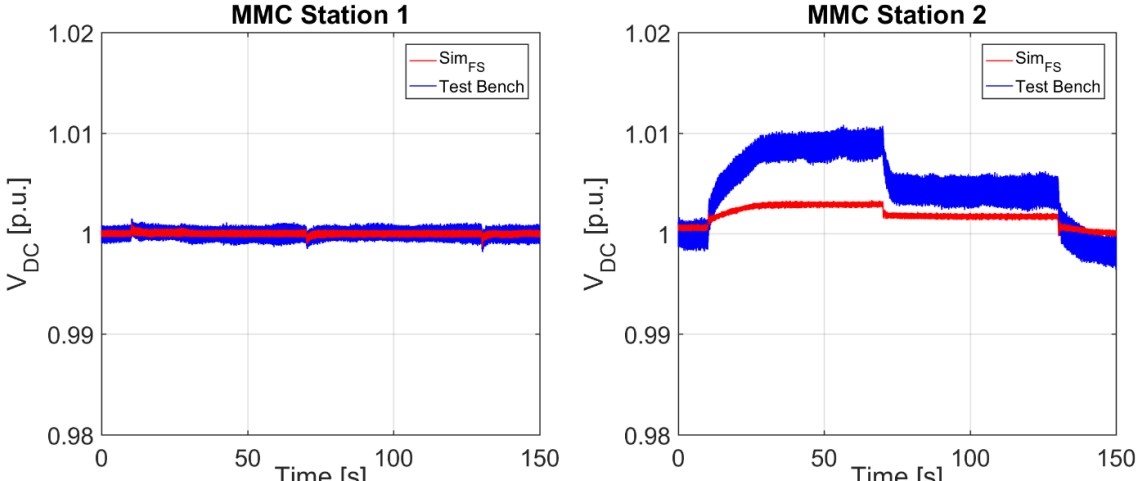

**Figure 8: DC voltage measured at MMC station 1 (left) and MMC station 2 (right)**

Figure 9 shows the DC current measured at the MMC stations. For the different wind speed changes at the OWF, a respective DC current can be observed. The DC current is defined as having a positive direction if it flows from the AC side to the DC

side of the MMC station. Correspondingly, at MMC station 2, for a wind speed $v_{wind} = 15$ m/s, a DC current of $I_{DC,MMC2} = 0.3$ p.u. can be observed. Apart from the ripples, the MMC Test Bench and the simulated full-scale model show a nearly identical behaviour with a slightly lower DC current level for the MMC Test Bench. Since the DC current level shows the same lower amplitude for the respective power infeed set-points, this difference can be related to the calibration of the PA. As for the DC voltage, this difference is considered acceptable. At MMC station 1, it can be seen that the simulated model and

the MMC Test Bench show the same behaviour as at MMC station 2, with the only difference that the DC current is negative because it flows from the DC grid to the AC grid.

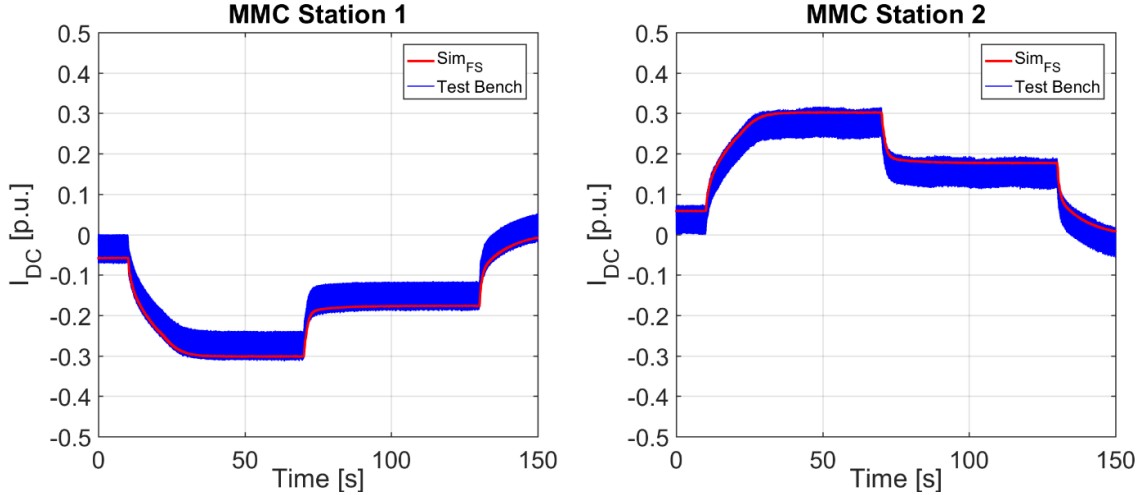

**Figure 9: DC current measured at MMC station 1 (left) and MMC station 2 (right)**



Figure 10 shows the active power measured at both terminals. At terminal 2, the active power transfer is determined by the
wind speed, to which the OWF is exposed. Initially the wind speed set-point is at $v_{wind}$ = 5 m/s and the corresponding active
power infeed of the OWF to the DC grid is $P_{MMC2}$ = 0.08 p.u. At 10 s, the wind speed increases to $v_{wind}$ = 15 m/s and at
terminal 2, an active power infeed of $P_{MMC2}$ = 0.3 p.u. can be observed when the active power reaches its steady state. The
simulated full-scale model and the MMC Test Bench show very close results with no significant difference. The difference
between the simulated model and the MMC Test Bench are the ripples of the active power signals. As for the other analysed
signals, the MMC Test Bench shows higher ripples compared to the full-scale model due to the difference in the number of
submodules. At 70 s, the wind speed changes to $v_{wind}$ = 12 m/s and the simulated model and the MMC Test Bench show the
same system behaviour as for the previous wind speed set-point. At terminal 1, the simulated full-scale model and the MMC
Test Bench show a similar qualitative system behaviour. However, the active power measured at terminal 1 has a lower
magnitude for the MMC Test Bench than for the full-scale model. This is assumed to be due to the higher losses of the DC
circuit of the MMC Test Bench. Further investigation regarding the PI-sections of the MMC Test Bench and their simulated
counterparts will be investigated. Nevertheless, this difference in the active power measured at terminal 1 is considered as
acceptable and further investigations in this regard are not presented in this paper.

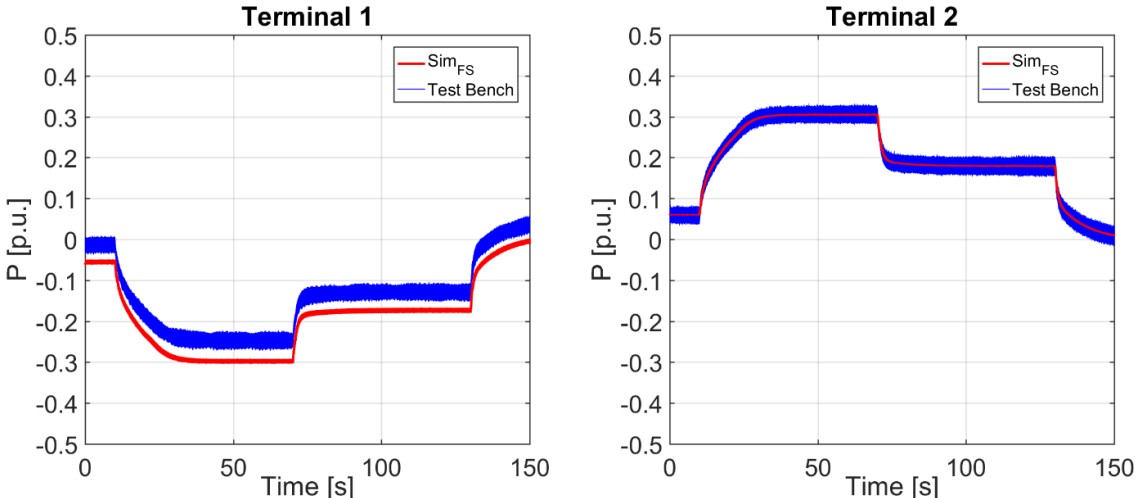

**Figure 10: Active power measured at terminal 1 (left) and at terminal 2 (right)**

**7 Discussion**

In this paper, the demonstration of a HVDC linked OWF using a lab-scaled PHiL setup, i.e. the MMC Test Bench is done.
The network configuration is initially simulated as a full-scale system. Then the point-to-point network configuration is scaled
down to the nominal values of the MMC Test Bench. The network is then implemented with the MMC Test Bench. MMC
station 1 of the network controls the DC voltage and MMC station 2 of the network is in islanded mode and thereby provides
the AC voltage. The OWF connected to the grid forming MMC station is exposed to different wind speed changes and



accordingly provides the active power to the DC grid. The AC voltage, the DC voltage and current and the active power measured at both MMC stations are compared and analysed for the different wind speed set-point changes.

The results show that the simulated full-scale model the MMC Test Bench show very close system behaviour with the only difference between the models being the higher signal ripples that are present for the MMC Test Bench. This difference arises

due to the fact that the MMCs of the MMC Test Bench consist of 10 SM per phase-arm compared to 350 SM per phase-arm for the full-scale model. Furthermore, the MMC Test Bench shows higher transmission line losses of the DC grid. However, for this paper they are considered as acceptable and are not investigated further. Overall, the MMC Test Bench shows very close system behaviour to the simulated models. This paper demonstrates the functionality of the MMC control algorithms including the islanded control of the MMCs. This investigation will serve as a basis for further investigations regarding multi-

terminal DC networks. Furthermore, the interoperability of the OWF and the MMCs, the black-start capability of the OWF as well as the validation of the equivalent frequency-based impedance models for the OWFs will be studied and validated.

## 8 Author contribution

F. Loku has implemented the converter control and the investigated network on the MMC Test Bench. P. Ruffing and C. Brantl have developed the MMC control algorithm for the simulated full-scale model. R. Puffer has supervised the investigations in

the paper.

## 9 Competing interests

The authors declare that they have no conflict of interest.

## 10 Acknowledgements

The authors' work has received funding from the European Union´s Horizon 2020 research and innovation program under
grant agreement No. 691714.

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
