# Peer review of "Demonstration of Offshore Wind Integration with an MMC Test Bench featuring Power-Hardware-in-the-Loop Simulation"

_Wind Energy Science, 2020_

## Referee Comment (RC1) · Anonymous Referee #1 · 13 Apr 2020

General comments

The articles topic is relevant to the journal, however, lack scientific rigours. The HiL implementation of MMC has been studied in many previous articles, with details about the challenges on HiL implementation and connection. As a discussion paper this doesn't goes beyond that level of technicality and in reviewer's opinion the contribution is not clear.

Technical comments

Another issue is the lack of demonstration with respect to dynamic cases (fault cases), which is of interest from the connection or operational point of view. Whether the

current state of the PHiL is sufficient enough to reflect the transient studies is not addressed. This is a short coming.

PHiL interface is touched upon with emphasis on used approach, however, the design criterion for filters, scaling methods etc could have been discussed in more detail. This could facilitate a clear discussion and understanding on the issues related HiL implementation when connecting devices operating at two voltage levels through a power amplifier.

Another question is related to the AC system implementation, it is not clear how the AC system is modelled in the RTS. Is it a source behind impedance or is it with multiple buses, please elaborate?

The schematic shown is Fig.3-5 are confusing, for instance, where is the DC network implemented hardware or RTS, similarly what is it inside the RTS which represents a wind turbine, what are the details of the WT test system inside RTS. A high-level hardware implementation with components representation could help the readers.

The authors also discuss on the ITM interface and its suitability for facilitating hardware connection to RTS, for strong grid from a stability point of view. The question will be how this interface performs for weak grid connection and what adjustments are needed to be considered to facilitate a stable PHiL connection.

The design of filters and associated delays in Interface algorithms is an important aspect to ensure stable HiL operation, as shown in Fig 6. However, detailed filter design for the demo experiment has been not taken into consideration.

The results for steady-state power flow models does not sufficiently demonstrates the HiL operation in reviewer's opinion. This is particularly interesting when the power flow is not reaching the full capacity, currently reaching inly at 0.3 p.u. 1/3 of the capacity of the device (see Fig 10). The comparison needs to be performed at 1 p.u. to see the differences between simulation and experiment and measurement errors.

As indicated earlier, more results especially on transient cases could of interest to scientific community.

The references to prior work are not sufficient for a topic of such interest. More academic papers should be included to identify the issues and solutions related to PHiL implementation (only one journal paper out of 16 is not sufficient enough.

---

## Referee Comment (RC2) · Anonymous Referee #2 · 13 Apr 2020

General comments:

I find that the focus and key contribution of the paper is not clear. Although HIL/PHIL results can be of great use, they are mainly for demonstration or validation purposes. PHIL results on their own are not of great interest – although I think the presented setup looks great and seems to behave reasonably, just making the HIL setup representative of a real system is not necessarily of significance (particularly for wind and HB-MMC systems which are now quite well understood). I would suggest that the focus of the paper should be more on the key outcome of the research – what is the thing that you are actually trying to validate using the PHIL setup – and then the PHIL results are

very nice to have but not the main focus. Otherwise, if the intention is to write a paper about how to design and configure a small scale demonstrator, then this could be of interest provided significant novel details are provided about the setup, the scaling of components, the other previously undocumented challenges when developing such a system. The title should then be adjusted accordingly so that it specifically reflects the key contribution.

I feel that you missed a key step in the design/demonstration process – I would expect that you also have a simulation only model of the low voltage system. Could it perhaps be of interest to discuss this model, and indicate if this model more closely matches the results from the full scale system or from the demonstrator? This could be an important step to identify where the differences in the results comes from. I suspect that you have already done this internally but I think it could be a useful addition to the paper (particularly when presenting the results for which the response of the full scale and the lab scale are not identical).

Detailed comments:

Page 2 Line 39 (P2 L39): The benefit of HIL/PHIL over pure simulation is critical to this paper. I think that the authors should provide more specific details here regarding the actual benefit of PHIL in this application.

P3 L75: Very little detail is provided here about the offshore wind farms, given that the paper claims to focus on the wind integration. I would expect more detail and more justification of the choice of wind farm – is the chosen case study typical and representative?

P5 L100: Although here there is some discussion of the implementation of the hardware, there are no details about the choices that were made when designing the system – e.g. how did you ensure that each component is representative of a full scale system? How are the MMC capacitors sized? How are the cable parameters chosen so that they are representative of a full scale cable? What phenomena is this system

representative for and what is it not representative for?

P9 L187: Here, and in other places in the paper, a start-up sequence is briefly mentioned but not detailed. Was this start-up sequence developed in this work? Or is it a standard sequence for which a reference should be provided?

Table 3: What is the basis for this wind speed case study – it looks like it would be reasonable to test a few power set points of the HVDC converter but this wind profile is surely not realistic for a real system. Might it be of interest to test a more realistic wind profile?

Fig8(right): The profile of the voltage is clearly very similar, but the magnitude is quite different. Could this be a problem with the scaling of the system? I suggest that the ('per unit') resistance of the cable is different between the full scale model and the lab scale demonstrator, and this causes the big difference in voltage for a given current. More discussion of the scaling would be useful to gain more insights into this point.

P10 L229: Following on from the point above I would certainly not say that the DC voltage is almost identical.

Fig10: I agree with "Anonymous Referee #1" that the 0.3pu power set point is perhaps not the most convincing one to demonstrate successful operation. I think that either a realistic case study should be presented (e.g. more detailed wind profile so that the full response of the controllers can be evaluated) or if you only wish to test the converters then you should demonstrate all points on the converter's PQ envelope to actually demonstrate that the converter can operate as specified. That said, I don't see too much added value in demonstrating the full PQ envelope of a converter that is perhaps not optimized (e.g. in terms of capacitance, current rating,. . .), and that for a converter topology which is well understood by academia and industry.

---

## Author Comment (AC1) · 11 May 2020

Dear Referee, Initially we would like to thank you for taking the time to revise our manuscript. The points mentioned by your side are very valid and helpful points which we are also considering in further investigations and development of our further research.

General Comments: Referee comment: The articles topic is relevant to the journal, however, lack scientific rigours. The HiL implementation of MMC has been studied in many previous articles, with details about the challenges on HiL implementation and connection. As a discussion paper this doesn't goes beyond that level of technicality

and in reviewer's opinion the contribution is not clear. Author's comment: While it is true that the HiL implementation of MMC has been studied in previous articles, according to the research conducted by the author a (P)HiL implementation of MMCs including the DC grid as a hardware, which is represented by Pi-sections has not been conducted yet. Additionally, in this article, the controllability of the hardware MMC stations as hardware components is demonstrated including a connected OWF. Furthermore, several studies have shown that depending on the configuration, Pi-sections can emulate a DC cable up to a certain frequency range especially for steady state cases. In this article this is investigated and is shown that for a complete hardware DC grid, additional parasitic resistances have to be included if the same dynamic response to a set-point change of the simulated full-scale model and the PHiL system is desired. Thereby, this paper shows that it may not be sufficient to derive the Pi-section parameters from an EMT cable model but further parasitic resistances must be considered as well. Technical comments Referee comment: Another issue is the lack of demonstration with respect to dynamic cases (fault cases), which is of interest from the connection or operational point of view. Whether the current state of the PHiL is sufficient enough to reflect the transient studies is not addressed. This is a short coming. Author's comment: Fault scenarios are out not part of the scope of the paper as initially it has to be made sure that the steady state behaviour of a system is identical to a respective simulation model. In order to properly reflect the transient studies with the described PHiL system, phenomena such as the parasitic resistances have to be addressed, such that they do not represent unrealistic losses for the simulated full-scale models. An approach to solve this would be to partially reduce the parasitic resistances in the PHiL setup and partially consider them when defining a DC grid configuration. Referee comment: PHiL interface is touched upon with emphasis on used approach, however, the design criterion for filters, scaling methods etc could have been discussed in more detail. This could facilitate a clear discussion and understanding on the issues related HiL implementation when connecting devices operating at two voltage levels through a power amplifier. Author's comment: The filters and scaling methods are explained

in more detail in section 4 after the reviewer's comments. Referee comment: Another question is related to the AC system implementation, it is not clear how the AC system is modelled in the RTS. Is it a source behind impedance or is it with multiple buses, please elaborate? Author's comment: The AC system is modelled as a single voltage source with a grid impedance in series. As it is modelled as a strong grid, from our view there was no need to represent it as multiple buses.

Referee comment: The schematic shown is Fig.3-5 are confusing, for instance, where is the DC network implemented hardware or RTS, similarly what is it inside the RTS which represents a wind turbine, what are the details of the WT test system inside RTS. A high-level hardware implementation with components representation could help the readers. Author's comment: The text explaining Fig 3 has been modified to emphasise that the DC grid is hardware, as well as the details describing the WT have been added to section 2 and 1 respectively. Referee comment: The authors also discuss on the ITM interface and its suitability for facilitating hardware connection to RTS, for strong grid from a stability point of view. The question will be how this interface performs for weak grid connection and what adjustments are needed to be considered to facilitate a stable PHiL connection. Author's comment: As for the transient studies, the weak grids in a PHiL setup are considered a topic on its own as detailed stability analysis and filter design methods would be required. This is unfortunately out of the paper's scope. However, this is a very valid point and an important research topic that will be investigated in the very near future. Referee comment: The design of filters and associated delays in Interface algorithms is an important aspect to ensure stable HiL operation, as shown in Fig 6. However, detailed filter design for the demo experiment has been not taken into consideration. Author's comment: For the used ITM method on the paper, a filter is used only for the voltage measurements, as the OWF is simulated as full scale model for the MMC Test Bench as well. The ripples present in the AC voltage that is provided by the islanded controlling MMC Station would otherwise be scaled up and lead to an unstable system. As the OWF is in full-scale, the ripples on the measurement current are not significant as well as their scaling down is not

considered significant. Therefore a filter is not needed for the measured current. A detailed filter design would definitely be of a great interest when weak grids are investigated as the effects of the filter on the stability of the interface algorithm for a weak grid are be of great importance. Referee comment: The results for steady-state power flow models does not sufficiently demonstrates the HiL operation in reviewer's opinion. This is particularly interesting when the power flow is not reaching the full capacity, currently reaching inly at 0.3 p.u. 1/3 of the capacity of the device (see Fig 10). The comparison needs to be performed at 1 p.u. to see the differences between simulation and experiment and measurement errors. Author's comment: A comparison between the simulation and the PHiL system at 1 p.u. power flow is definitely of great interest. However, this would mean that the interface algorithm has to be adjusted for a weak grid and this is out of the paper's scope. According to our view, a power rating of 1 p.u. is not unconditionally needed for this paper, as this paper' aim is to demonstrate the challenges when designing a PHiL system with real hardware MMC stations and with real hardware DC grid, such as the additional parasitic resistances in real hardware components, which influence the dynamic response of the system. These effects must be studied prior implementing weak grids or fault scenarios. Weak grids as well as the fault scenarios would be topics on their own and need detailed investigations and further studies regarding the stability of the interface algorithms. Referee comment: The references to prior work are not sufficient for a topic of such interest. More academic papers should be included to identify the issues and solutions related to PHiL implementation (only one journal paper out of 16 is not sufficient enough. Author's comment: The references are updated to include additional work done on P(HiL) systems.

---

## Author Comment (AC2) · 11 May 2020

Dear Referee, Initially we would like to thank you for taking the time to revise our manuscript. The points mentioned by your side are very valid and helpful points which we are considering in the further investigations and development of our further research.

General Comments: Referee comment: I find that the focus and key contribution of the paper is not clear. Although HIL/PHIL results can be of great use, they are mainly for demonstration or validation purposes. PHIL results on their own are not of great interest – although I think the presented setup looks great and seems to behave reasonably, just making the HIL setup representative of a real system is not necessarily of significance (particularly for wind and HB-MMC systems which are now quite well understood). I would suggest that the focus of the paper should be more on the key outcome of the research – what is the thing that you are actually trying to validate using the PHIL setup – and then the PHIL results are very nice to have but not the main focus. Otherwise, if the intention is to write a paper about how to design and configure a small scale demonstrator, then this could be of interest provided significant novel details are provided about the setup, the scaling of components, the other previously undocumented challenges when developing such a system. The title should then be adjusted accordingly so that it specifically reflects the key contribution. Author's comment: Thank you very much for your positive assessment of the paper and the very constructive comments. The main focus of the paper is to describe how to design a small scale PHiL system as well as to present the challenges and compare it to a full scale system. The title of the paper is adjusted accordingly. Referee comment: I feel that you missed a key step in the design/demonstration process – I would expect that you also have a simulation only model of the low voltage system. Could it perhaps be of interest to discuss this model, and indicate if this model more closely matches the results from the full scale system or from the demonstrator? This could be an important step to identify where the differences in the results comes from. I suspect that you have already done this internally but I think it could be a useful addition to the paper (particularly when presenting the results for which the response of the full scale and the lab scale are not identical). Author's comment: The results from the simulation model of the low voltage system were already done and are now included in the paper as well. As the model of the low voltage system showed the exact same behaviour as the full-scale model it was initially omitted. However, after the referee's comments it is included in the paper and is used to show the effects that can make a difference between a simulated model and the PHiL system, such as parasitic resistances, which after are considered can lead to a big improvement in the results. Detailed Comments: Referee comment: Page 2 Line 39 (P2 L39): The benefit of HIL/PHIL over pure simulation is critical to this paper. I think that the authors should provide more specific details here regarding the actual benefit of PHIL in this application. Author's comment: The benefits of the HIL/PHIL in the paper have been elaborated in the mentioned section. Referee comment: P3 L75: Very little detail is provided here about the offshore wind farms, given that the paper claims to focus on the wind integration. I would expect more detail and more justification of the choice of wind farm – is the chosen case study typical and representative? Author's comment: The aim of the paper is to describe how to demonstrate a HVDC connected OWF with a small scale demonstrator and compare it to a full-scale model, by identifying and analysing the differences between the simulation model and the PHIL system. In order to make a direct comparison between the simulated models and the PHIL system, the same simulation circuits, such as the AC systems and OWF have to be used. As no faults are investigated in the paper, an average model of OWF is used, where equivalent voltage sources represent the voltage source converters of the OWF. This model is assumed as sufficient to investigate the control system dynamics and the system behaviour. Furthermore, the simulation time step of such a system is typically ts = 50 $\mu$s, which makes it possible to use exactly the same model for the offline simulations and a PHiL system. The details are added to the mentioned section.

Referee comment: P5 L100: Although here there is some discussion of the implementation of the hardware, there are no details about the choices that were made when designing the system – e.g. how did you ensure that each component is representative of a full scale system? How are the MMC capacitors sized? How are the cable parameters chosen so that they are representative of a full scale cable? What phenomena is this system representative for and what is it not representative for? Author's comment: The aim of this work is to compare and show the differences between a simulated full-scale system of a HVDC connected OWF with a developed PHIL system consisting of already tested lab-scaled MMC converters. The DC grid, i.e. the Pi-sections were designed initially for a full-scale system. The Pi-Section parameters are extracted from a model of a submarine +-320 kV cable designed in PSCAD/EMTDC. The Pi-section

parameters are then scaled down by considering the voltage and the power levels on the DC side of the simulated full-scale system and the PHiL system respectively. As it is discussed in the paper, the results of the simulated full-scale model and the simulated lab-scale model show very close match. In order for the MMC Test Bench to show a close match to the simulated models, further parasitic elements have to be considered in the simulated models. Further details are added to the mentioned section. Referee comment: P9 L187: Here, and in other places in the paper, a start-up sequence is briefly mentioned but not detailed. Was this start-up sequence developed in this work? Or is it a standard sequence for which a reference should be provided? Author's comment: The mentioned start-up is developed in this work, considering the start-up sequence of the hardware MMC stations and the (current) controlling mode of the power amplifiers. Referee comment: Table 3: What is the basis for this wind speed case study – it looks like it would be reasonable to test a few power set points of the HVDC converter but this wind profile is surely not realistic for a real system. Might it be of interest to test a more realistic wind profile? Author's comment: Additional power set points have been added to the results. The wind profile used in this work is only representative, as it serves its purpose for the investigations carried out in this paper. However, a realistic wind profile is definitely of a great interest along with a more detailed model of an offshore wind farm and they are planned to be tested in future work.

Referee comment: Fig8(right): The profile of the voltage is clearly very similar, but the magnitude is quite different. Could this be a problem with the scaling of the system? I suggest that the ('per unit') resistance of the cable is different between the full scale model and the lab scale demonstrator, and this causes the big difference in voltage for a given current. More discussion of the scaling would be useful to gain more insights into this point. Author's comment: This is a very valid point. In order to explain this point in more detail, a simulated lab-scale model is added to the paper. The results show that the simulated full-scale model and the simulated lab-scale model show identical behaviour and identical voltage level. However, when the MMC Test Bench is considered, the magnitude of the voltage is higher. This comes due to the additional parasitic resistance on the DC side, which results from the additional losses coming from the cable connecting the MMC Stations with the Pi-sections. If this parasitic resistance is measured, scaled up and added to the simulated full-scale model, then the behaviour of the MMC Test Bench and the full-scale model is very close. This is explained in more detail in the mentioned section in the paper. Referee comment: P10 L229: Following on from the point above I would certainly not say that the DC voltage is almost identical Author's comment: This remark has been corrected in the paper.

Referee comment: Fig10: I agree with "Anonymous Referee #1" that the 0.3pu power set point is perhaps not the most convincing one to demonstrate successful operation. I think that either a realistic case study should be presented (e.g. more detailed wind profile so that the full response of the controllers can be evaluated) or if you only wish to test the converters then you should demonstrate all points on the converter's PQ envelope to actually demonstrate that the converter can operate as specified. That said, I don't see too much added value in demonstrating the full PQ envelope of a converter that is perhaps not optimized (e.g. in terms of capacitance, current rating,. . .), and that for a converter topology which is well understood by academia and industry. Author's comment: A comparison between the simulation and the PHiL system at 1 p.u. power flow is definitely of great interest. However, this would mean that the interface algorithm has to be adjusted for a weak grid and this is out of the paper's scope. According to our view, a power rating of 1 p.u. is not unconditionally needed for this paper, as this paper' aim is to demonstrate the challenges when designing a PHiL system with real hardware MMC stations with real hardware DC grid, such as the additional parasitic resistances in real hardware components, which influence the dynamic response of the system. These effects must be studied prior implementing weak grids or fault scenarios. Following the reviewer's comment, additional wind set-point changes have been added to the results.